# Structural basis for substrate recognition mechanism of human SLC26A7

Xiaorong Li[1,2,3,5], Xiaoxu Yang[2,3,5], Xiaoli Lu [2,3,5], Bingqian Lin[2,3,5], Yuanyuan Zhang[1,2,3,5], Bangdong Huang[2,3], Yutong Zhou[2,3], Jing Huang [2,3]✉, Kun Wu[2,3]✉, Qiang Zhou [2,3]✉ & Ximin Chi [4]✉

Solute carrier family 26 (SLC26) mainly mediates transmembrane transport of various anion ions, including chloride and other halide ions, bicarbonate, oxalate, and sulfate. Many severe hereditary human diseases are correlated with SLC26 protein mutations. Here we report cryo-EM structures of human SLC26A7 in apo and iodide binding states. We identify non-canonical binding site for halide ions in SLC26A7. Molecular dynamics simulation and electrophysiological assay confirm the functional importance of key residues involved in iodide and chloride coordination. Together, our discovery marks a step towards an in-depth understanding of SLC26 family protein transport mechanisms.

Anion homeostasis is crucial for maintaining water-electrolyte balance in organisms. Transmembrane transport of anions including chloride, iodide, sulfate, bicarbonate, oxalate is of significance for kidney stone formation[1], epithelial cell fluid excretion[2], sperm post-testicular maturation and capacitation[3], sulfate uptake by chondrocytes[4], and iodide storage and thyroid hormone synthesis[5]. Dysregulation of anion transport will lead to various metabolism and development disorders containing congenital goitrous hypothyroidism[5], diastrophic dysplasia[4], congenital chloride-losing diarrhea[6], and hearing loss[7]. The altered oxalate or bicarbonate excretion in intestine and kidney is closely related to gastric-induced injury in duodenal mucosa, hyperoxaluria, hyperuricemia and kidney stone formation[8].

SLC26 family contains 11 genes in human, encodes 10 proteins that mainly mediate transmembrane transport of various anion ions, including chloride, bicarbonate, formate, oxalate, nitrate, sulfate, and hydroxide[9]. SLC26 family can be roughly divided into three clans based on their substrate preference: sulfate transporter (SLC26A1, SLC26A2), Cl⁻/HCO₃⁻ exchangers (SLC26A3, SLC26A4, and SLC26A6), and Cl⁻ channels (SLC26A7, SLC26A9)[2]. SLC26A7 is not permeable to bicarbonate[10,11]. It is located on the basolateral membrane of outer medullary collecting duct in kidney[12] and basolateral membrane of gastric parietal cells in stomach[13]. Some reports support the Cl⁻/HCO₃⁻

exchange activity in kidney[12,14,15]. It was also detected in the basolateral membrane of Reissner's cell in cochlea and exhibited chloride, NO₃⁻, and iodide channel activity[16]. Besides, the thiocyanate channel activity of SLC26A7 was reported in retinal pigment epithelium[17]. SLC26A7 and Pendrin (also known as SLC26A4) are demonstrated to transport iodide in thyroid[5,18]. Double knock-out *Slc26a7* in mice results in severe growth failure compared with *Slc26a4* knock-out groups in low iodide environment[19]. Mutations in *SLC26A7* were identified in patients with congenital goitrous hypothyroidism[5,20], indicating the close correlation between SLC26A7 function and iodide intake of thyroid. More diseases are related to function abnormality of SLC26 proteins or mutations in SLC26 protein genes, including nephrolithiasis[21], osteochondrodysplasia[22], Pendred syndrome[23], and deafness[24,25].

Structural study of SLC26 family proteins has made a profound process with the advantage of cryo-EM (cryo-electron microscopy) revolution. The structures of mammalian SLC26 proteins were determined after bacterial homolog SLC26Dg[26], which includes mouse Slc26a9[27], mouse and human SLC26A2[28,29], human SLC26A9[30], human SLC26A6[31], human Pendrin[32], dolphin[33], gerbil[34], human[35] and thermostabilized Prestin[36] etc. The structure information provides insights of SLC26 transport and regulation mechanisms. However, the substrate recognition and transport mechanism of SLC26s are not fully

[1]College of Life Sciences, Zhejiang University, Hangzhou, Zhejiang, China. [2]Research Center for Industries of the Future, Zhejiang Key Laboratory of Structural Biology, School of Life Sciences, Westlake University, Westlake Laboratory of Life Sciences and Biomedicine, Hangzhou, Zhejiang, China. [3]Institute of Biology, Westlake Institute for Advanced Study, Hangzhou, Zhejiang, China. [4]State Key Laboratory for Cellular Stress Biology, Innovation Centre for Cell Signaling Network, School of Life Sciences, Xiamen University, Fujian, China. [5]These authors contributed equally: Xiaorong Li, Xiaoxu Yang, Xiaoli Lu, Bingqian Lin, Yuanyuan Zhang. ✉e-mail: huangjing@westlake.edu.cn; riddywoo@hotmail.com; zhouqiang@westlake.edu.cn; chiximin@xmu.edu.cn

understood. Although SLC26A7 and Pendrin behave iodide transport activity, the iodide recognition and transport are still unknown. The transport path in SLC26 family could be more complicated as multiple paths can exist between the large interface of gate and core domains[37]. Identification of the substrate recognition mechanism would facilitate development of potential targeting molecules. In order to understand the substrate recognition and transport mechanism of SLC26s, we apply cryo-EM to solve high-resolution structures of human SLC26A7 in apo and iodide-loaded states at resolution of 3.2 and 3.1 Å respectively. The unique domain connection of SLC26A7 provides specified crosstalk between sulfate transporter and anti-sigma factor antagonist (STAS) domain and transmembrane domains (TMDs). We identify a substrate recognition site in SLC26A7 located in the pocket formed by TM5, TM13 and TM14. Ion binding stability and free energy along the path is analyzed by molecular dynamics simulation. Key residues in ion coordination are verified by electrophysiological assay. Conformation classification of SLC26 proteins reveal that the structure of SLC26A7 is captured in intermediate state with subtle core domain deformations. Our study paves path to thorough comprehension of SLC26s transport mechanisms as well as pharmaceutical development for SLC26-related diseases.

## Results

### Overall structures and domain architecture of SLC26A7

We solved the cryo-EM structure of SLC26A7 in apo state (SLC26A7_apo) with overall resolution of 3.2 Å and core region resolution of 3.0 Å (Fig. 1a, b). The details of protein purification structure determination can be found in Methods (Supplementary Fig. 1 and Supplementary Table 1). The cryo-EM map quality allows model-building of most protein sequences (Supplementary Fig. 2). SLC26A7 is a domain-swapping dimer, in which the STAS domain of one protomer is located near the transmembrane domain of the other protomer. Several cholesterol-like molecules are presented around the TMDs (Supplementary Fig. 3a), which is similar to the structures of human Prestin, highlighting a conserved regulation from lipid bilayer in SLC26 proteins[35]. The TMD can be divided into core domain (TM1-4, TM8-11) and gate domain (TM5-7, TM12-14) (Fig. 1b, c). TM3 and TM10 contain unwounded regions that form a canonical substrate binding pocket which is conserved in UraA-fold transporters. TM1-7 and the TM8-14 are pseudo-symmetrical to each other (Fig. 1d).

The N-terminal region located in the bottom of STAS domain is stable in SLC26A9, Pendrin, Prestin and SLC26A6 structures, but flexible in SLC26A7. A highly conserved motif YXXXR (Supplementary Fig. 3b)[38] provides structure integrity by cation-pi interaction between the tyrosine and arginine. STAS domain stays rigid upon N-terminal sequence binding (Supplementary Fig. 3c). Marginal area of SLC26A7 STAS domain is invisible in cryo-EM map due to structural flexibility, which includes intervening sequence (IVS) region and β1-β2 loop (Supplementary Fig. 3d). An interface between STAS domain and TMD is formed by Y619 in STAS domain α3 and N432' in TM12'. An extensive hydrophobic interaction network involving α3, α2 and TM12' further enhances the interdomain and interprotomer connection (Supplementary Fig. 3e). Another interprotomer interface conserved in SLC26s is located around the end of TM14, consisting of multiple hydrophobic residues from TM14 and α1-β1 loop (Supplementary Fig. 3f).

Electrophysiological assays were reported to be suitable for evaluation of SLC26A7 transport activity[14]. Both chloride and iodide transport were measured by whole cell patch-clamp recording technique. SLC26A7 exhibits chloride and iodide transport activity, and the chloride current is sensitive to DIDS (4,4'-diisothiocya- nostilbene-2,2'-disulfonic acid) as previously reported[10]. No significant transport capability difference was detected in SLC26A7 with respect to chloride and iodide (Fig. 1e, Supplementary Fig. 3g). The Cl⁻ and I⁻ transport activity of SLC26A7 was further confirmed by ion gradient assay.

Currents rise with elevated ion concentration in the bath buffer. No significant difference was found in Cl⁻ and I⁻ transport under the same ion concentration (Supplementary Fig. 3h).

### Iodide coordination in SLC26A7

To understand the mechanisms of substrate recognition and transport by SLC26 members, sodium iodide was added in purification of SLC26A7 (Supplementary Fig. 1). The cryo-EM map of SLC26A7 with iodide (SLC26A7_I) shows additional non-protein densities in the long cleft between core and gate domain when compared with SLC26A7 apo state (Fig. 2a). The range of SLC26A7_I map (Fig. 2a, middle) was adjusted and subtracted by SLC26A7_apo (Fig. 2a, left) map to obtain difference map (Fig. 2a, right), which supports the binding of iodide ions at these sites, namely I1 and I2, respectively (Fig. 2a). I1 locates in the canonical substrate binding pocket of SLC26 family proteins (I1 site thereafter), which is coordinated by TM3 and TM10. The residues F68, A365, F104, and A105 form the binding pocket of I1. The coordination is established by the phenyl group of F104, F68, and the amide nitrogen of A105 (Fig. 2b, Supplementary Fig. 2c). Structure comparison of SLC26s around I1 site reveals that F104 is highly conserved (Fig. 2c, Supplementary Fig. 4a). A water molecule is also found in the vicinity, providing solvent interface of I1 (Fig. 2a, b, Supplementary Fig. 2c). Different from the canonical substrate binding pocket, iodide ion in another binding site (I2 site) is mainly coordinated by residues from gate domain, which includes A186 and H189 in TM5, and A452 and A453 in the loop between TM13 and TM14. L109 in TM3 from core domain is also involved in the coordination of I2 (Fig. 2b, Supplementary Fig. 2d). Structural and sequence alignment of SLC26 homologs indicates that there are mostly hydrophobic residues at the site of L109, and hydrophilic residues at the site of H189, such as histidine, glutamine or threonine (Fig. 2c, Supplementary Fig. 4a). AlphaFold 3[39] prediction results also point out the possibility of chloride binding around I2 site (Supplementary Fig. 4b).

Molecular dynamics (MD) simulations of SLC26A7 dimer was applied to further investigate the ion binding stability at these sites. The simulation was initialized with Cl⁻ or I⁻ bound at both sites, respectively. Both Cl⁻ and I⁻ ions are stable at I1 site during 100 ns MD simulations. Nevertheless, Cl⁻ ion at I2 site quickly dissociates into the outward solvent environment, whereas the I⁻ ion stays longer. Therefore, I⁻ is stable in I2 site but Cl⁻ is instable here. This suggests the cryo-EM density observed at I2 site more likely to be I⁻ (Fig. 3a).

Patch-clamp was then applied to verify the roles of residues involved in ion-coordination. Various mutants at F104 or H189 were designed. The whole-cell currents were recorded for wild type (WT) and the mutant SLC26A7 in the extracellular solution containing Cl⁻ or I⁻. The Cl⁻ currents were increased in cells expressing F104A or F104K mutant and decreased in F104E mutant. As for I2 site, the chloride transport activity of the H189A or H189E mutant was inhibited (Fig. 3b, Supplementary Fig. 4c). Similar phenomena were observed for I⁻ transport (Fig. 3b, Supplementary Fig. 4d). These results highlight the roles of these residues for ion permeation.

### Conformational change in SLC26 transport cycle

The boom of SLC26s structure study facilitates in-depth conformational state analysis[27,28,32,33,35,40,41]. The transmembrane region of the experimental structures of SLC26s are paired compared, generating a structural atlas of SLC26 family (Fig. 4a). In asymmetric structures, the conformation of each protomer is considered separately[32]. Three major groups are identified in this structural atlas, which are inward-facing, intermediate and outward-facing groups. Bacterial homolog SLC26Dg shows difference with other groups. The outward-facing group contains only 4 members of SLC26A4 with high structural similarity within the group. The inward-facing group contains 17

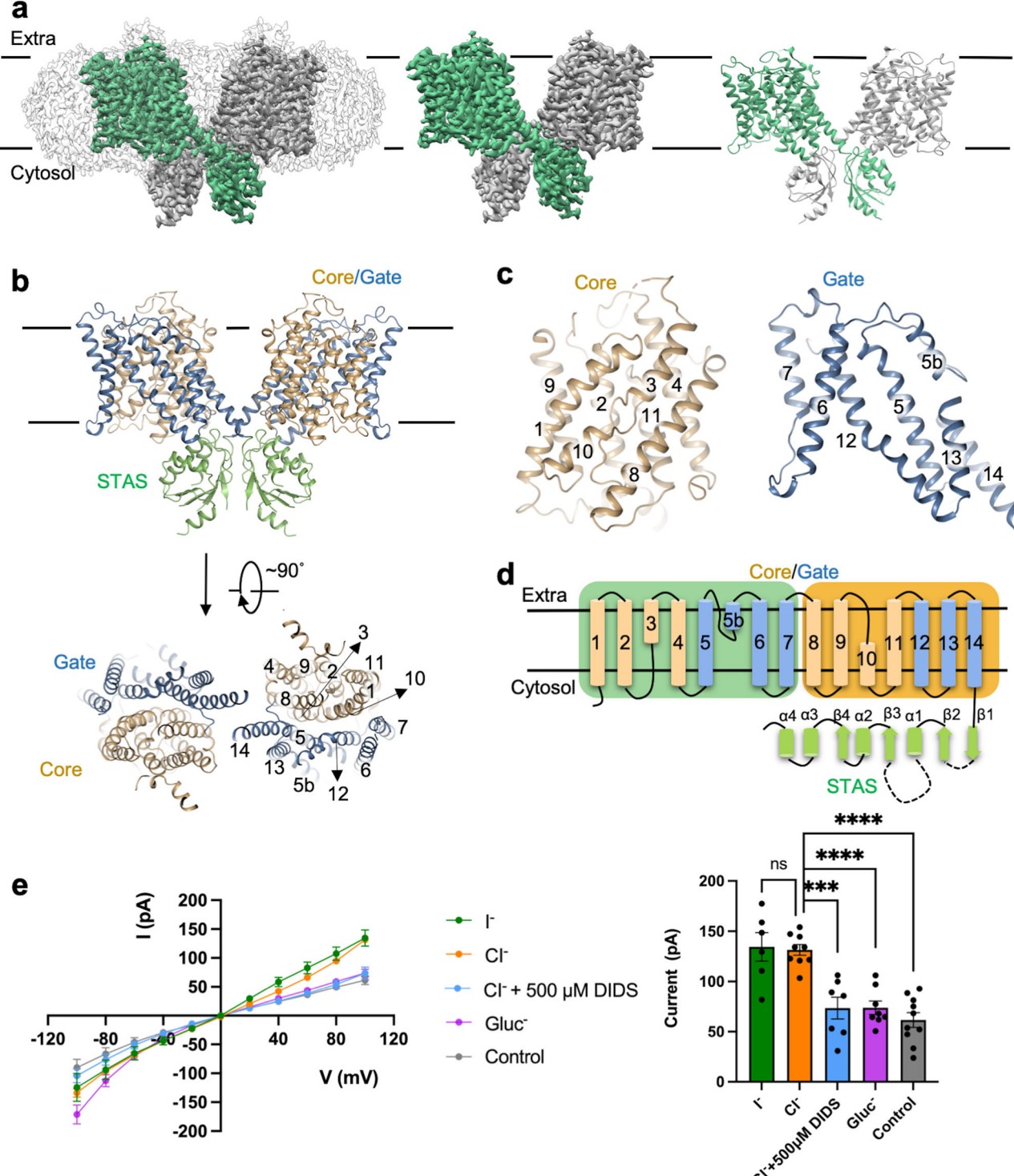

**Fig. 1 | Overall structure of human SLC26A7 and functional verification. a** Cryo-EM reconstruction and model of SLC26A7. Different protomers are colored in gray and green. **b** Structure model of SLC26A7 colored by different domains. Rotated view below shows the transection of transmembrane domain. **c** Enlarged view of core and gate domain. The numbers of transmembrane helices are labeled. **d** A cartoon scheme of SLC26A7. The pseudosymmetry in transmembrane helices are highlighted by green and orange background rectangles. **e** SLC26A7 behaves Cl⁻ and I⁻ transport activity. HEK293T cells transfected with SLC26A7 were used to measure the I/V curves of I⁻ and Cl⁻. The Cl⁻ currents inhibited by DIDS, Gluc⁻ and

Control currents were also recorded by incubating cells in corresponding bath solutions. The currents at a voltage of +100 mV were shown in the right. The number of cells patched for I⁻, Cl⁻, Cl⁻ + 500 μM DIDS, Gluc⁻, and Control were $n = 6$, 9, 7, 8, 10, respectively. Data represents the means ± SEM, \*\*$P < 0.01$, \*\*\*$P < 0.001$, \*\*\*\*$P < 0.0001$ (The significance is calculated by one-way ANOVA with Dunnett's test. The exact $P$ values between each comparison are Cl⁻ vs. I⁻: 0.9979, Cl⁻ vs. Cl⁻ + 500 μM DIDS: 0.0001, Cl⁻ vs. Gluc⁻: < 0.0001, Cl⁻ vs. Control: <0.0001. SEM is short for standard error of the mean).

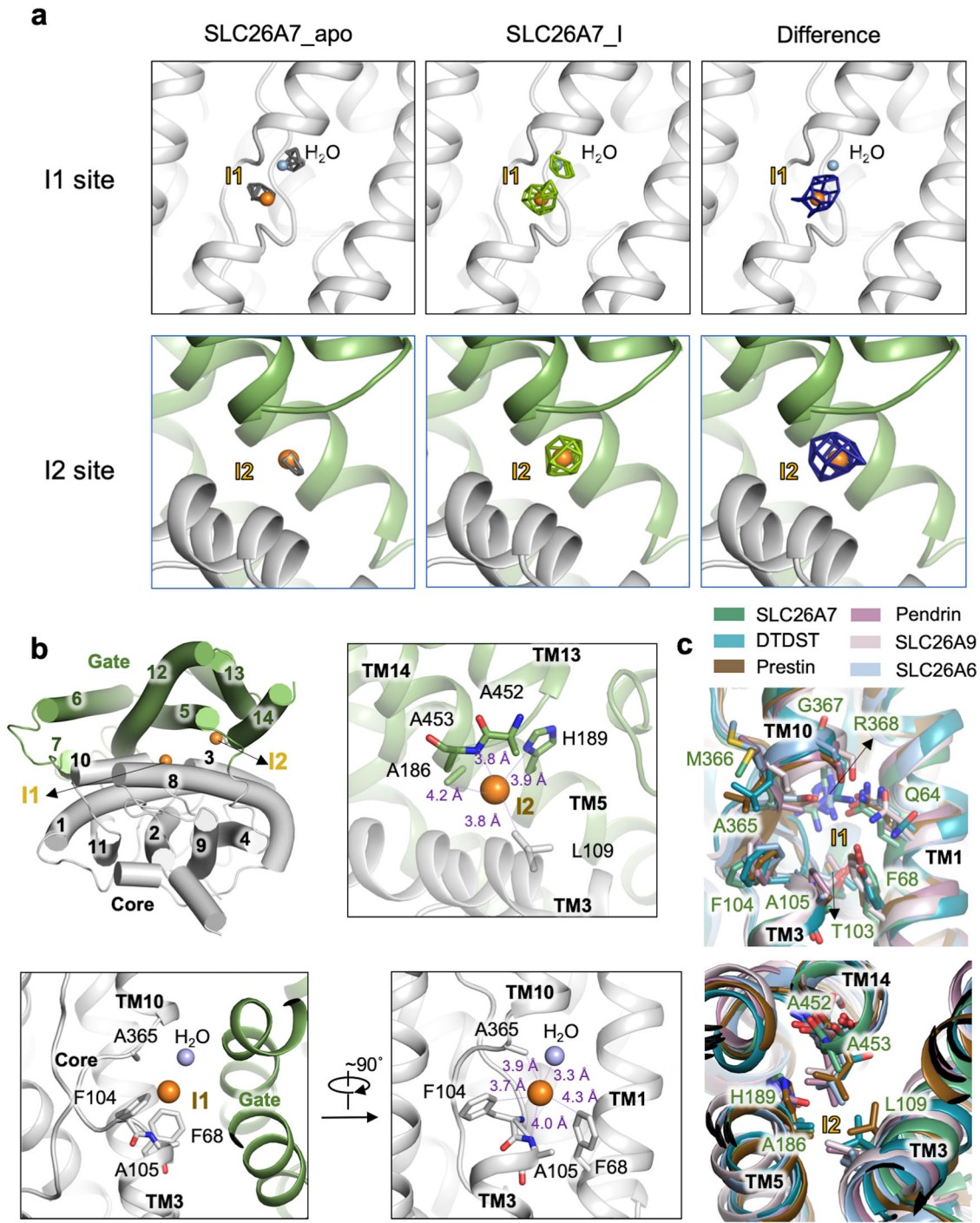

**Fig. 2 | Structure of human SLC26A7 reveals two iodide binding sites. a** Model of SLC26A7 with iodide loaded is docked to cryo-EM map of SLC26A7 apo (gray), SLC26A7_I (green) and difference map (deep blue), respectively. Additional density in I1 and I2 site are clear in SLC26A7_I map. Iodide ions are built in these places. The maps are shown at σ = 5. For I2 site in SLC26A7 apo map, σ = 3. **b** Two iodide ions bind in the cleft of core and gate domains of SLC26A7. Iodide ion in I1 site (canonical substrate binding site) is stabilized through interactions with residues from core domain, including F104, A365 and F68. Amide nitrogen of A105 also partici-pate in ion coordination. Iodide ion in I2 site is coordinated mainly by residues from gate domain, including H189, L109, A186, and mainchain of A452 and A453. **c** Structure alignment of SLC26A7_apo, Pendrin (PDB 7WK1), Prestin (PDB 7LGU), SLC26A6 (PDB 8OPQ), SLC26A9 (PDB: 7CH1), and DTDST (PDB 7XLM) around I1 and I2 site. Key residues involved in I1 and I2 ion binding are shown. only residues in SLC26A7 are labeled.

members which behave limited diversity. As for the intermediate group, 23 members are included and exhibit the highest heterogeneity due to structure variations of the extracellular loop of TM3-TM4. The structures of SLC26A7 in apo and iodide loading states are alike with RMSD being 0.427 Å inferring substrate binding induces little con-formational change. Both SLC26A7 structures in our study are classi-fied in the intermediate group.

We further analyze the conformation of SLC26A7 by struc-ture alignment with representative structures of inward-facing (SLC26A9, PDB 7CH1)[30], intermediate (Prestin, PDB 7LH2)[35], and outward-facing (Pendrin, PDB 7WLE)[32]. Gate domains were aligned to compare the conformational change of core domain. The RMSD between Prestin and SLC26A7 is 2.5 Å (Fig. 4b), while the RMSD raise to ~6 Å in comparison with SLC26A9 (Fig. 4c) and

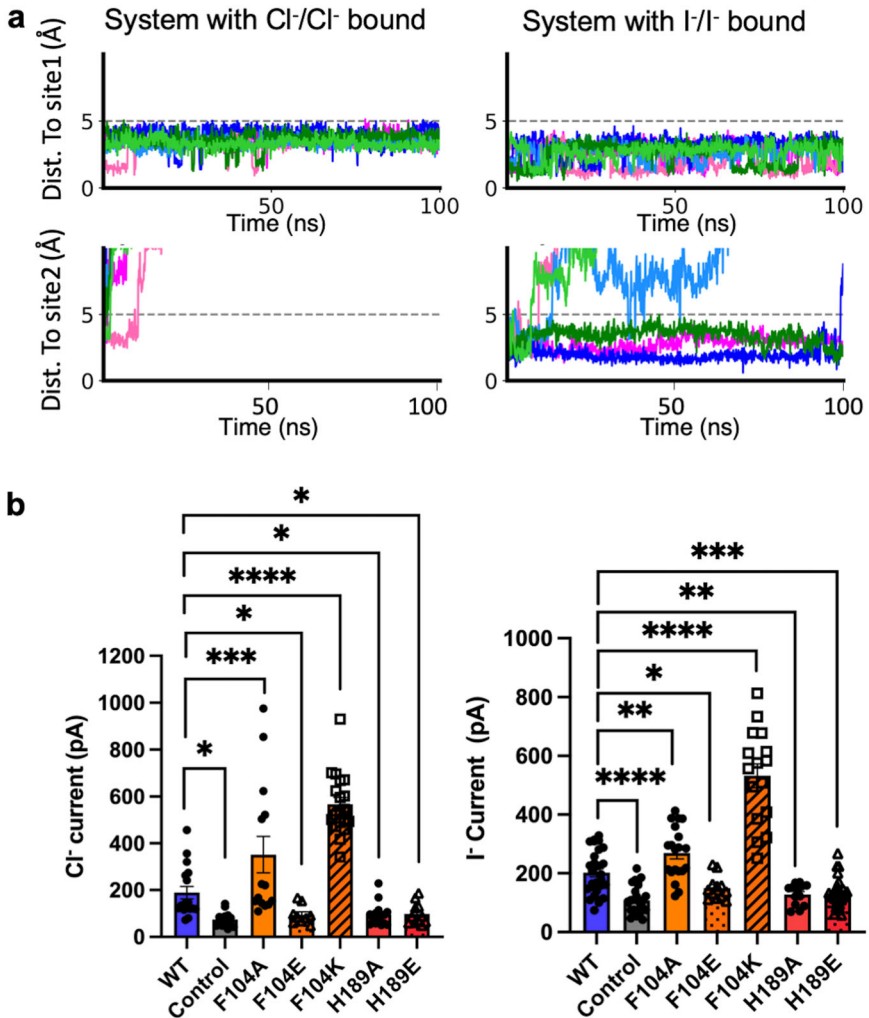

**Fig. 3 | Verification of ion species and key residues in coordination. a** MD simulation estimating the ion species and binding stability in I1 and I2 sites indicates that while both Cl⁻ and I⁻ are stable in I1 site, I2 site is more stable for I⁻ binding and instable for Cl⁻. In each panel, the six different colored lines represent the distance between ion and each site in the dimer system across different replicate simulations. The simulations consisted of three independent 100 ns replicates, with snapshots of the system saved every 100 ps for analysis. **b** The Cl⁻ current and I⁻ current (at +100 mV) of WT, control and mutants were shown respectively. The number of cells patched for each group (in the order of WT, Control, F104A, F104E,

F104K, H189A, H189E) were $n = 17, 14, 14, 9, 19, 16, 11$ (Cl⁻ current) and $n = 29, 26, 20, 12, 16, 13, 27$ (I⁻ current). Data represent the means ± SEM, *$P < 0.1$, **$P < 0.01$, ***$P < 0.001$, ****$P < 0.0001$ (The significance is calculated by one-way ANOVA with uncorrected Fisher's LSD test. The exact $P$ values for Cl⁻ currents measurements are: WT vs. Control: 0.0132, WT vs. F104A: 0.0005, WT vs. F104E: 0.0721, WT vs. F104K: <0.0001, WT vs. H189A: 0.0355, WT vs. H189E: 0.0679. The exact $P$ values for I⁻ currents measurements are: WT vs. Control: <0.0001, WT vs. F104A: 0.0019, WT vs. F104E: 0.0400, WT vs. F104K: <0.0001, WT vs. H189A: 0.0030, WT vs. H189E: 0.0004).

Pendrin (Fig. 4d). The highest similarity with Prestin confirms that the conformation of SLC26A7 captured in this study is intermediate state. The conformational state of SLC26A7 can be summarized in Fig. 4e. Although there are some varieties within the same conformation, the overall movement of core domain from cytosolic to extracellular side supports the elevator transport model of SLC26 family proteins (Supplementary movie 1).

The core domain deformations are obvious when compared with Prestin (intermediate, Fig. 4b), especially in TM3, TM10, and TM8. When aligned with other Prestin intermediate structures and up state structures, residue conformational change exists when zoomed in substrate binding site. Displacement of M366 (L397 in Prestin) interferes I1 binding (Supplementary Fig. 4e). When aligned with Prestin down states, SLC26A7 moves towards extracellular side. Meanwhile, substrate binding site detects large displacement, such as movements in Q64, F104, and T103 (Supplementary Fig. 4f). Therefore, deformation of core domain in

SLC26s can provide subtle adjustment of substrate recognition and coordination during transport cycles.

## Discussion
In this study, we have determined the cryo-EM structures of human SLC26A7 in apo and iodide loading states. Whole cell patch clamp recording supports that SLC26A7 transports Cl⁻ and I⁻ similarly (Fig. 1e). Apart from the canonical substrate binding site (I1 site) between TM3 and TM10 half helices of the core domain, another iodide binding site (I2 site) was found located in the TM5 and the TM13/TM14 connecting loop (Fig. 2). The alignment of I1 site indicates the high conservation of F104 in TM3 and Q64 in TM1 (named after residues of SLC26A7, thereafter). F104A enhanced the Cl⁻ and I⁻ transport activity of SLC26A7. This indicates that F104 is an energy barrier in ion transduction, which can act as a possible substrate selection site. Consistent with this hypothesis, F104E, of which the substitution to glutamic acid enhanced the negative charge of this site,

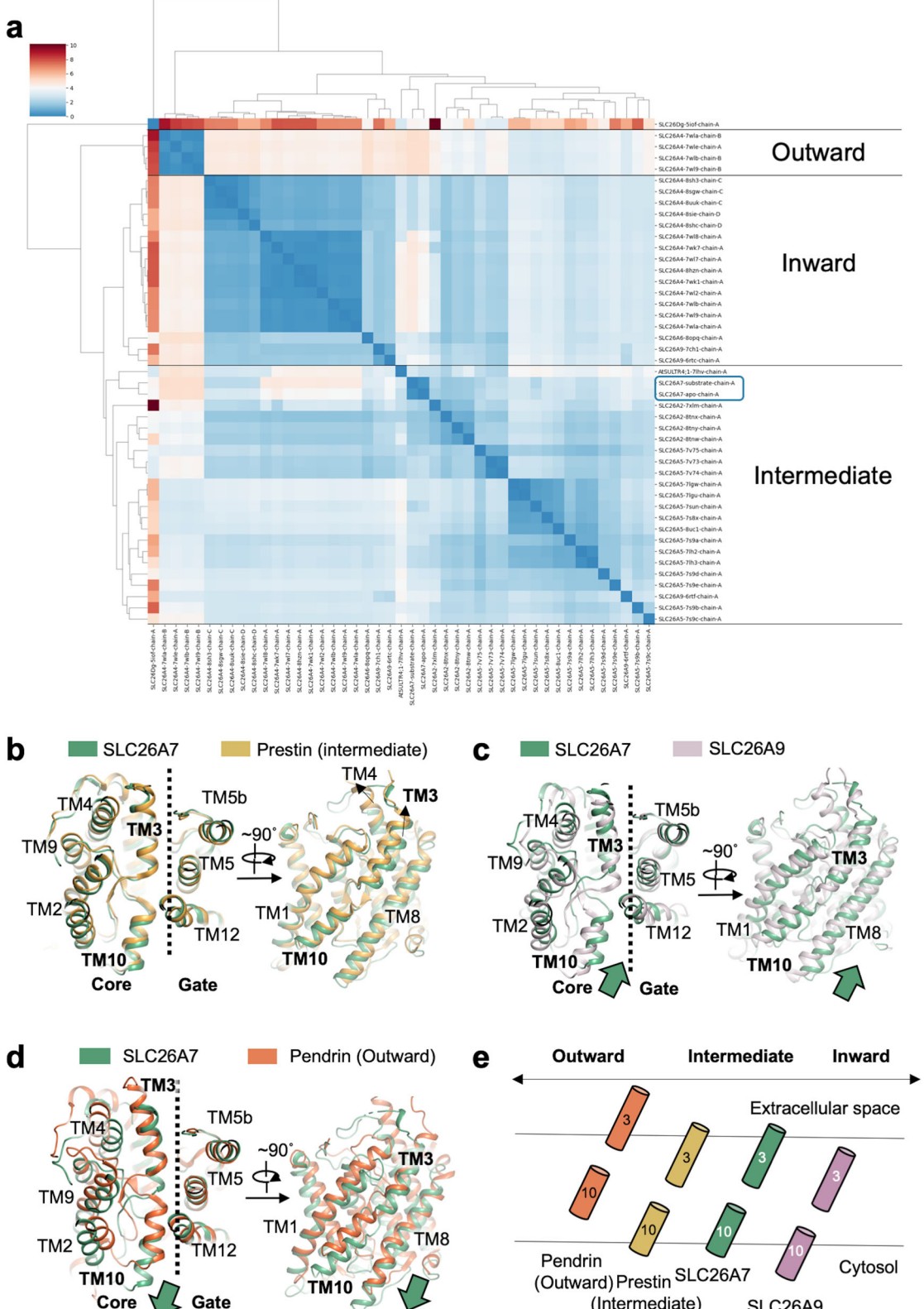

**Fig. 4 | Conformation comparison of SLC26A7 to other SLC26 homologs.**
**a** Cluster analysis of SLC26s structures. Paired RMSD of experimental structures of SLC26s was calculated, based on which a heatmap was generated. Three major groups representing inward-facing, intermediate and outward-facing can be identified by color differences. The color codes correspond to RMSD (unit: Angstrom). SLC26A7 apo and substrate (iodide) binding state (highlighted by blue box) are classified in intermediate group. **b**–**d** Structure alignment of SLC26A7 with SLC26A9 (Inward-facing, PDB 7CH1), Prestin (intermediate, PDB 7LH2), and Pendrin (Outward-facing, PDB 7WLE) respectively. Gate domains are aligned to show core domain conformational change. Conformation of SLC26A7 core domain is well-aligned to Prestin (intermediate). **e** Cartoon scheme to present the structure alignment results in a simpler way. Only TM3 and TM10 are depicted.

inhibited $Cl^-$ passage. The anticipated upregulation of $Cl^-$ and $I^-$ transport are observed for the F104K mutant (Fig. 3b, Supplementary Fig. 4c, d). The ion in I2 site is mainly coordinated by residues from gate domain. Structure comparison of I2 site indicates high conservation of A186, hydrophobic and hydrophilic features of L109 and H189 respectively. AlphfaFold 3 prediction of chloride bound SLC26A7 structure also confirms the existence of the second ion coordination (Supplementary Fig. 4b). Electrophysiological assay points out that H189 is indispensable for $Cl^-$ and $I^-$ transport, as both H189A and H189E inhibit the current strongly (Fig. 3b, Supplementary Fig. 4c, d). Overall, most residues involved in substrate recognition lack strict sequence conservation, which provides structure basis for the substrate variety. Substrate preference in SLC26s could therefore result from local environments.

The iodide ions were built in the SLC26A7_I model based on the following clues. Firstly, cryo-EM map comparison indicates strong additional signals in I1 and I2 sites of SLC26A7_I map (Fig. 2a, Supplementary Fig. 2c, d). The only difference in protein sample preparation is sodium iodide addition. It is iodide binding that induces the signal enhancement. Secondary, SLC26A7_apo shows weak density of suspected $Cl^-$ in I1 and I2 sites when the map threshold is lowered down, which indicates $Cl^-$ binding. But it is not enough for a robust model-building. Iodide ion has higher atom weight and exhibit better signal in cryo-EM data than chloride ion. Therefore, iodide ion provides rationality for the enhanced map density. Thirdly, MD simulation results support that while both $Cl^-$ and $I^-$ are stable in I1 site, only $I^-$ is stable in I2 site (Fig. 3a). Yet the binding affinity of $Cl^-$ and $I^-$ to SLC26A7 needs to be measured for ion species determination.

Core domain deformation is observed when compare SLC26A7 with other structures of SLC26s. Explicit transmembrane helixe migration can be detected when conformation changes (Fig. 4b). In similar conformation, deformation of residue side-chains are also presented, with potential influence of substrate binding (Supplementary Fig. 4e, f). In addition to the adjustment of core domain deformation, regulation of ion transport also comes from cytosolic side, which is conducted through interactions between STAS and TMD' in SLC26A7. When SLC26A7 and SLC26A9 are aligned to TM14, STAS of SLC26A7 moves towards cytosolic side. Meanwhile the gate domain from the other protomer moves towards cytosolic side correspondently (Supplementary Fig. 4g), highlighting the coupling between STAS and TMD'.

Corresponding to the two ion binding sites, two possible ion permeation pores can be proposed in SLC26A7 using HOLE[42], namely PORE1 and PORE2 respectively (Supplementary Fig. 5a–c). Four hydrophobic residues, namely L109, V190, M420, and M366, form a hydrophobic fence to separate the different pores (Supplementary Fig. 5b). The residues along PORE1 and PORE2 are highly conserved (Supplementary Fig. 5 d). Substrates in I1 are enclosed from both sides of the membrane, with restriction site radius less than 2 Å (Supplementary Fig. 5c), which is smaller than hydrated chloride ion radius[43]. Thus, it concludes the occluded substrates binding in I1 site, which is in consist with MD simulation results indicating stable chloride or iodide ion binding in I1 site (Fig. 3a). As for PORE2, similar situation is observed as ion in I2 is enclosed from both sides (Supplementary Fig. 5c). Residues along PORE1 and PORE2 are mostly conserved indicating functional importance. To quantitatively investigate the free energy of $I^-$ and $Cl^-$ along the pores, the umbrella sampling was performed to calculate the potential of mean force (PMF) along the two transport paths[44]. The results indicate that $Cl^-$ or $I^-$ is hard to escape once trapped at the I1 site due to the sharp energy barrier (Supplementary Fig. 5e), which is in consist with pore radius analysis. The I2 site is a global minimal for $I^-$ binding in PORE2, but only a local minimal for $Cl^-$. It is in consist with pore radius analysis, as I2 site is enclosed from both ends. However, it is easier for $Cl^-$ to escape from I2 site and diffuse into to extracellular space. Besides, the PMF at

outward region is lower than that at inward region, suggesting that both $Cl^-$ and $I^-$ are easier to enter from outward direction (Supplementary Fig. 5f). Together, our study further enriches the structure landscape of SLC26s with potential applications in specific drug development of related diseases.

## Methods

### Protein expression and purification

The full-length wild-type human *SLC26A7* (also known as SUT2, UniProt ID: Q8TE54) gene were cloned into a pCAG vector with an N-terminal flag tag (DYKDDDDK) and strep tag (WSHPQFEK). SLC26A7 was expressed in HEK293F (Invitrogen). Cells were cultured in SMM 293-TII (Sino Biological Inc.) under 5% $CO_2$ and 37 °C provided by a Multitron-Pro shaker (INFORS, 130 rpm). Transfection was carried out when the cells reached a density of ~$2.0 \times 10^6$ cells/mL. Approximately 1.5 mg plasmids and 3 mg polyethylenimine (PEI, Polysciences, MW 25000) were applied to each liter of cells. After 48 h of culture, cells were harvested in lysis buffer containing 20 mM HEPES (pH7.4) and 150 mM NaCl by centrifugation at 3800 g for 10 min.

Then the cells were lysed with protein inhibitors (1 μg/mL pepstatin, 1.3 μg/mL aprotinin, 5 μg/mL leupeptin, and 0.2 mM phenylmethyl sulfonyl fluoride (PMSF)) and detergent mixture which is comprised of 1% n-dodecyl-β-D-maltopyranoside (DDM, Anatrace) and 0.2% cholesteryl hemisuccinate (CHS, Anatrace) at 4 °C for 2 h. After centrifugation at ~$25,000\,g$ for 1 h, the supernatant was applied to anti-Flag M2 affinity resin (Sigma) at 4 °C. The resin was washed with buffer A (20 mM HEPES, pH7.4, 150 mM NaCl, 0.02% GDN) and eluted in the same buffer with 0.4 mg/mL Flag peptide. The elution was then loaded to Strep Tactin XT Resin and washed with buffer A, eluted in the same buffer with 10 mM desthiobiotin. Then the protein was further purified by size exclusion chromatography (SEC, Superose 6 Increase 10/300 GL, GE Healthcare) in buffer: 25 mM Tris, pH8.0, 150 mM NaCl, and 0.02% GDN. The peak fraction was analyzed by SDS-PAGE before being concentrated for apo state SLC26A7 samples. For obtaining SLC26A7 with iodide loading samples, sodium iodide was added to a final concentration of 20 mM throughout the protein purification process.

### Electron microscopy sample preparation and data collection

SEC purified protein solution was concentrated to 5–10 mg/ml, applied to Quantifoil Au 1.2/1.3 grids, and blotted for 3 s at 100% humidity and 8 °C on a Vitrobot before being plunge frozen in liquid ethane cooled by liquid nitrogen.

Data were collected in a 300 kV Titan Krios, which is equipped with GIF Quantum energy filter and a Gatan K3 detector. AutoEMation[45] was applied in automatically collection of super-resolution movie stacks with a defocus ranging from −2.2 μm to −1.2 μm. The nominal magnification is 81,000× and the pixel size is 1.087 or 1.077 Å/pixel. Each movie stack was exposed for 2.56 s in 32 frames with the total dose rate approximately 50 e⁻/Å². MotionCor2[46] was used in motion correction, during which dose weighting[47] is performed. Gctf[48] was applied in defocus values estimation.

Relion 3[49] was applied in automatically Particles-picking, 2D classification, 3D classification and auto-refinement. The particles automatically picked were subjected to several rounds of 2D classifications. Selected particles were then subjected to global angular searching 3D classification against an initial model generated from map of SLC26A9[30]. Selected particles in 3D classification were subjected to 3D auto-refinement. Local defocus refinement and protein mask were applied in improving the resolution. The resolution was estimated with the gold-standard Fourier shell correlation 0.143 criterion[50,51] with high-resolution noise substitution[52]. Supplementary Fig. 1 provides details of the data collection and processing. The corresponding data and refinement statistics are included in Supplementary Table 1.

## Model building

The initial model of SLC26A7 with iodide (SLC26A7_I) was built by MDFF[53] with the reported SLC26A9 structure (PDB ID: 7CH1)[30]. Apo SLC26A7 were modified based on SLC26A7_I model. PHENIX[54] and COOT[55] was applied to modify the model. The dimer structure model was obtained by C2 symmetry through PHENIX. Secondary structure and geometry restraints were set in structure refinement with PHENIX. To monitor the overfitting of the model, the model was refined against one of the two independent half maps from the gold-standard 3D refinement approach. All structure figures were prepared using PyMol[56], UCSF Chimera[57] and UCSF ChimeraX[58].

## Molecular dynamics simulation

The simulation systems with $Cl^-$ bound or $I^-$ bound were set based on the cryo-EM structures, protonation states were predicted by PROPKA[59]. The CHARMM-GUI was used to embed the dimer structure in a membrane containing POPC lipids, and solvated into TIP3 waters and neutralized with 150 mM NaCl[60,61]. The final constructed system contained approximately 169,000 atoms with size of $119 \times 119 \times 129$ Å$^3$. All production simulations were performed using OpenMM package[62] patched with the PLUMED 2.5 plugin[63].

The CHARMM36m force field was used for proteins[64]. The force field parameters for $Cl^-$ are directly adopted from the CHARMM force field[65]. For $I^-$, the parameters were derived based on the reported difference in hydration free energy between $I^-$ and $Cl^-$ ions[66]. The parameters for $Cl^-$ are Rmin/2 = 2.27 and $\varepsilon = -0.15$, while those for $I^-$ are Rmin/2 = 2.76 and $\varepsilon = -0.184$. The simulation temperature was set to 310 K, and the pressure was maintained at 1 atm. Periodic boundary conditions and particle-mesh Ewald summation for electrostatic calculation were used throughout the whole simulations. Monte Carlo membrane barostat was employed to keep the dimension in X-Y direction and Z direction was set free to move. Then two systems were firstly equilibrated for 1 ns with 1 fs time-step and 10 ns with 2 fs time-step with gradual decrease of constraints on the heavy atoms. To assess the ion binding stability using the conventional MD simulations[67–69], the 100 ns production simulations with three replicas were performed to sample the snapshots every 100 ps.

To quantitatively investigate the binding properties of $I^-$ and $Cl^-$ at I1 and I2 sites, we performed the umbrella sampling to construct potential of mean force (PMF) along two transport pathways[44]. Specifically, the collective variables were defined as the central path along the pore, calculated using the HOLE program[42]. HOLE identifies the pore's central axis by determining the pathway of maximum radius within the channel, ensuring that it accurately reflects the ion conduction pathway. Then, two transport paths for $Cl^-$ and $I^-$ were obtained to define the reaction coordinate for umbrella sampling simulations. For each window in umbrella sampling, the position of ion was restrained using a harmonic potential with force constant $k = 5$ kJ mol$^{-1}$ Å$^{-2}$. In total, 34 windows for 100 ns were calculated for $Cl^-$ and $I^-$ transport from PORE1 and PORE2, respectively, which amounted to 13.6 μs MD simulations. The PMF was then calculated with WHAM (http://membrane.urmc.rochester.edu/?page_id=126). All the trajectories analysis were performed using VMD package[70] and some in-house scripts.

## Electrophysiology

HEK293T cells were grown in DMEM media (Gibco, C11995500BT) supplemented with 10% FBS (Sigma, F8318) at 37 °C and 5% $CO_2$, in Cell Culture Flask T-25. Using lipofectamine 3000 (Thermo Fisher, L3000001), 2 μg plasmid DNA was transfected to cells. After 24 h, the whole cell recording of the patch clamp technique was used to measure the $Cl^-$ and $I^-$ currents in GFP (control), mutants and SLC26A7 (WT) transfected HEK293T cells.

The pipette solution contained (mM): 140 N-methyl-D-glucamine-Cl, 1 $MgCl_2$, 2 EGTA, 5 ATP, and 10 HEPES (pH7.3 with Tris). For SLC26A7 wild type transport activity assay, "$I^-$"group was measured in the $I^-$ bath solution contained (mM): 145 NaI, 10 HEPES, 10 Glucose, 1 $CaI_2$, 1 $MgI_2$ (pH7.3 with NaOH). "$Cl^-$"group was measured in the $Cl^-$ bath solution contained (mM): 145 NaCl, 10 HEPES, 10 Glucose, 1 $CaCl_2$, 1 $MgCl_2$ (pH7.3 with NaOH). 500 μM chloride channel inhibitor DIDS was added in $Cl^-$ bath buffer for group of "$Cl^-$ + 500 μM DIDS". "Control" group was measured in $Cl^-$ free bath solution prepared by replacing $Cl^-$ with gluconate. To evaluate the key residues involved in ion binding, $Cl^-$ currents of SLC26A7 wild type and mutants were recorded in the $Cl^-$ bath solution, while $I^-$ currents were recorded in bath buffer which substituted NaCl by NaI. For ion gradient transport assay, the bath buffers for "149 mM" groups were the same as the ones in SLC26A7 wild type transport activity assay. In "50 mM" groups, [$I^-$] or [$Cl^-$] was reduced to 50 mM, and 99 mM sodium gluconate was added. An agar salt bridge perfused with 3 M KCl was used for reference electrode when recording the iodide currents.

The current was recorded using an Axopatch 200B amplifier (Molecular Devices) and digitized with a Digidata 1550B converter (Molecular Devices), filtered at 5 kHz, sampled at 10 kHz. Currents were measured by a standard protocol that stepped the membrane potential from a holding potential of 0 mV for 50 ms to membrane potentials between −100 and +100 mV at 20 mV steps for 200 ms with 1 s intervals between sweeps. Current recording and analysis were performed with the pClamp 11.2.2 software and GraphPad Prism 9.5 (GraphPad Software).

## Reporting summary

Further information on research design is available in the Nature Portfolio Reporting Summary linked to this article.

## Data availability

The cryo-EM maps have been deposited in the Electron Microscopy Data Bank (EMDB) under accession codes EMD-60662 (SLC26A7 in apo state) and EMD-60660 (SLC26A7 in iodide binding state). The atomic coordinates have been deposited in the Protein Data Bank (PDB) under accession codes 9IKX (SLC26A7 in apo state) and 9IKV (SLC26A7 in iodide binding state). The initial data and PLUMED input files required in MD simulations to reproduce the results reported in this paper are available on PLUMED-NEST, the public repository of the PLUMED consortium, as plumID:24.034. Source data are provided with this paper.

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

## Acknowledgements
The work was supported by National Science Foundation of China grant 32100975 and grant 32471304 to Ximin Chi, 82241081 to Qiang Zhou.

## Author contributions
Xiaorong Li, Yuanyuan Zhang and Yutong Zhou performed the plamids construction, protein purifications and sample preparations. Yuanyuan Zhang acquired the cryo-EM images from Titan Krios. Xiaoxu Yang and Bingqian Lin performed the electrophysiology experiments. Kun Wu provided insightful suggestions and facilitated the electrophysiology experiments design. Xiaoli Lu performed the molecular dynamic simulations and umbrella sampling. Jing Huang supervised the molecular dynamic process. Bangdong Huang performed the structure pair-wise analysis. Ximin Chi processed the Cryo-EM data, prepare the figures and wrote the manuscript. Qiang Zhou provided precious suggestions in Cryo-EM data processing. Jing Huang, Kun Wu, Qiang Zhou, and Ximin Chi analyze the data and modified the manuscripts. All authors discussed the results, critically read the manuscript, and approved the manuscript for submission. Ximin Chi supervised this work.

## Competing interests
The authors declare no competing interests.
