## [Transparent Peer Review file · Nature Communications]

Structural basis for substrate recognition mechanism of human SLC26A7

Corresponding Author: Professor Ximin Chi

Version 0:

Reviewer comments:

Reviewer #1

(Remarks to the Author)

In this manuscript, Li et al. described the cryo-EM structure of human SLC26A7 in both the apo and I⁻ binding states. The authors revealed important residues for substrate binding and confirmed their significance using patch clamp experiments. The novelty of this work lies in the claim of a second substrate binding site (I₂), which sits in a second potential translocation pathway in the transporter. However, there are some significant discrepancies in the finding that needs to be addressed. Please see my comments below:

Major Concerns:

1. From the electrophysiology data, it is clear that there is no difference in transport efficiency for Cl⁻ and I⁻. Since the authors never discussed SLC26A7's binding affinities to Cl⁻ and I⁻, one would assume them to be comparable. If that's true, then it is hard to imagine that under 150mM NaCl, the apo cryo-EM structure has no Cl⁻ binding, while under 150mM NaCl+20mM NaI, the I⁻ binding structure has two clearly visible I⁻. In previous structural studies of SLC members, it seems quite common to find Cl⁻ densities.
2. The authors claim that there are two separate transport pathways, with each passing through one substrate binding site. This is a bold claim as, at least in SLC26 members, no such observation has been made. It is a consensus note that SLC26 members go through transport cycles under the classic alternating access mechanism. The observed 2nd I⁻ binding site is most likely resides in the extracellular side of the only substrate translocation pathway.
3. The authors claim that the determined cryo-EM structures of SLC26A7 are both in the intermediate states. Are we talking about transporters or channels? From my understanding, it is mostly a transporter, but with some Cl⁻ channel activity (PMID: 32119864). If it's in the intermediate state, is the substrate binding pocket (between TM3 and TM10) accessible from both sides of the membrane? Such an intermediate state in transporters is also referred to as an occluded state that is close on both ends of the translocation pathway. The authors need to clarify the status of the structure more clearly.
4. What is the point of Fig. 4a? The authors performed such cluster analysis by including both experimental and predicted structures, which does not make too much sense. One compares either all available experimental models or only computationally predicted models.

Minor concerns:

1. Numerous grammar errors need to be carefully corrected.
2. Page 2: known SLC26 structures are not complete. Human SLC26A2 is missing. It's probably better to include structures of determined bacterial homologs, too.
3. The model building can be further improved. For example, the backbone of multiple residues is clearly wrong, and the Ramachandran plot can be improved.
4. Supp fig 2e legend is incorrect.

Reviewer #2

(Remarks to the Author)

The manuscript by Li and colleagues reported the structures of human SLC26A7 in apo and iodide binding states. Like the structure of Pendrin, the structure of SLC26A7 forms a dimer and adopts an UraA-fold. The TMD can be divided into gate and core domains. Based on structural analysis, two putative ion binding sites were identified. One canonical ion binding

site is located in the pocket formed by TM3 and TM10 highly which is conserved in UraA-folded transporters. What is potentially new is that the iodide-loaded structure also potentially revealed a non-canonical ion binding site, which is located in the pocket formed by TM5, TM13 and TM14 in the gate domain. Furthermore, key residues involved in ion coordination were studied using electrophysiological assays. Additional evidence is required as outlined in my comments to support several of the conclusions.

Major

1) In line 90-91, the authors stated that "Several cholesterol-like molecules are presented around the TMDs (Supplementary Fig. 2a)". The densities are in the peripheral regions, where the resolution is low. How can the authors distinguish the densities are not part of the detergent micelle.

2) In line 119-121, the authors said "The cryo-EM map of SLC26A7 with iodide shows additional non-protein densities in the long cleft between core and gate domain when compared with SLC26A7 apo state. Iodide ions were therefore built in these sites, namely I1 and I2, respectively". In addition, a water molecule was identified in I1 site. However, the quality of the map was not shown. The authors should calculate the difference map. The omitted density can be used to evaluate the signal of the built ions.

3) In line 155, the authors claimed that "Both pores share the same entry and exist". But shown in Fig.3b, pore1 and pore2 have a different entry and exit. This needs clarification.

4) In line 240-242, "minor deformation of the core domain provides subtle adjustment of substrate recognition and coordination during transport cycles.". The statement needs more evidence to support it, to help address in greater detail the structural detail/changes in the transport cycle.

5) Although the authors showed that density observed at I2 site is likely to be I- by MD, there is no proof that the density at I1 is I-. It could also be Cl-. The possibility of endogenous Cl- binding during protein expression and purification cannot be excluded.

6) In some UraA-folded transporters, TM8 is also part of the canonical binding pocket. Are there any residues located in TM8 involved in ion coordination in SLC26A7?

Minor

1) In Fig.1, add labels to transmembrane helices of the model.

2) In line 230-231, AlphaFold 3 prediction of chloride bound SLC26A7 was performed. What was the AlphaFold 3 prediction of iodide bound SLC26A7 since iodide was built in I2 in the model?

3) Add SEC profile and gel for sample purification in Supplementary Fig.1. Specify the buffer conditions used for SEC in protein expression and purification section.

Reviewer #3

(Remarks to the Author)

Version 1:

Reviewer comments:

Reviewer #2

(Remarks to the Author)

I have carefully reviewed the revised manuscript and the authors responses. I have no further comments.

Reviewer #3

(Remarks to the Author)

Reviewer #4

(Remarks to the Author)

This is a technical review for the MD simulations:

- 1) The phrase "Simulations was carried out every 100 ps with duration of 100 ns with 3 replications" is unclear and needs clarification and editing. Specifically, what do the authors mean by "Simulations was carried out every 100 ps"?
- 2) In Figure 3a, it would be helpful if the authors could explain what each line, represented in different colors, stands for.
- 3) Can the authors mention and cite any molecular dynamics (MD) work where a time length of 100 ns was used to assess the structural stability of ion binding sites in channel proteins?
- 4) The authors should clarify which parameters were used for the I- and Cl- ions. The cited reference in the Supplementary Material points to the parametrization of cations (supplementary Ref. 23).
- 5) Related to Figure 3a, which site is the most prone to ion instability? The colored lines should perhaps report on this. Moreover, could it be possible to have a system with Cl-/I- bound?
- 6) Detailed information should be provided on how HOLE was used to build the path collective variable (CV) mapped with Umbrella Sampling.
- 7) Error bars should be included for the potential of mean force (PMF) profiles in Supplementary Figure 4.
- 8) The X-axis label of the PMF (Supplementary Figure 4e,f), which simply reports "in" and "out," should indicate which window each portion of the PMF belongs to. Ideally, for comparison with the tunnel features shown in Supplementary Figure 4c, the free energy profile should also be plotted as a function of the "Distance along the permeation Path."
- 9) It should be clarified which portion of the PMF corresponds to the "ion binding" site. Again, the X-axis of the PMF should report the distance along the permeation path and the corresponding Umbrella sampling window. This will facilitate following the final part of the manuscript's discussion.
- 10) References should be provided when citing Umbrella Sampling.
- 11) For the sake of clarity and to allow the community to build on the simulations performed by the authors, PLUMED input files and corresponding simulation input files should be shared via the PLUMED-NEST.
- 12) The manuscript, including the main text and the Methods section in the Supplementary Material, would benefit from careful proofreading and editing.

Version 2:

Reviewer comments:

Reviewer #2

(Remarks to the Author)

I have carefully reviewed the revised manuscript and the authors responses. I have no further comments.

Reviewer #3

(Remarks to the Author)

Reviewer #4

(Remarks to the Author)

I have carefully read the revised manuscript, and the authors have sufficiently addressed my concerns. I only have a final suggestion regarding the wording of the sentence in the legend of Figure 3: "The simulations consisted of three independent 100 ns replicates, with conformational sampling performed every 100 ps."

I assume the authors intend to convey that sampling was continuous throughout the 100 ns simulations, with snapshots of the system saved at 100 ps intervals. However, the current phrasing — "with conformational sampling performed every 100 ps" — could suggest that sampling only occurred at those specific intervals, rather than continuously.

To avoid this potential confusion, I suggest rephrasing it to something like: "with snapshots of the system saved every 100 ps for analysis."

This would clarify that data points were recorded at regular intervals.

Point-by-point response to the reviewers' comments

REVIEWER COMMENTS

Reviewer #1 (Remarks to the Author):

In this manuscript, Li et al. described the cryo-EM structure of human SLC26A7 in both the apo and I⁻ binding states. The authors revealed important residues for substrate binding and confirmed their significance using patch clamp experiments. The novelty of this work lies in the claim of a second substrate binding site (I2), which sits in a second potential translocation pathway in the transporter. However, there are some significant discrepancies in the finding that needs to be addressed. Please see my comments below:

Thank you for reviewing our manuscripts. The point-to-point answers are attached to each of the questions below.

Major Concerns:

1. From the electrophysiology data, it is clear that there is no difference in transport efficiency for Cl⁻ and I⁻. Since the authors never discussed SLC26A7's binding affinities to Cl⁻ and I⁻, one would assume them to be comparable. If that's true, then it is hard to imagine that under 150 mM NaCl, the apo cryo-EM structure has no Cl⁻ binding, while under 150 mM NaCl + 20 mM NaI, the I⁻ binding structure has two clearly visible I⁻. In previous structural studies of SLC members, it seems quite common to find Cl⁻ densities.

Thank you for pointing out this question. In structures of Prestin (PDB ID: 7LGU), Pendrin (PDB ID: 7WK1), and SLC26A2 (PDB ID: 8TNW) observe chloride binding in the canonical substrate binding site, which is I1 site in SLC26A7. It is also noticed that chloride ions are absent in the structures of Prestin (PDB ID: 7S8X) and SLC26A9 (PDB: 6RTC and PDB: 7CH1), although all of these structures are solved under 150-360 mM NaCl (PMID: 34695838 and 32818062, 31339488). Thus, chloride binding is not always presented in SLC26s structures.

In the purification process, the only difference between SLC26A7_I and SLC26A7 apo is the addition of sodium iodide. I⁻ also has larger atomic weight and thus stronger signal in cryo-EM data processing than Cl⁻. We have checked the cryo-EM maps of SLC26A7_I and SLC26A7_apo respectively. When the maps are shown in the comparable contour level, obvious non-protein density is presented in SLC26A7_I map (Figure. 1, grey and blue maps). We calculated the difference map of SLC26A7_I and SLC26A7 apo. The density for iodide binding is clear in the difference map (Figure. 2). Moreover, MD simulation also points out that while Cl⁻ and I⁻ are stable in I1 site, I⁻ is more stable in I2 site. Considering all clues above, we conclude iodide ions binding in I1 and I2 sites in the cryo-EM structure of SLC26A7_I. Please refer to the updated manuscripts for the panel of difference map and corresponding paragraphs.

Figure 1| Cryo-EM density map (SLC26A7_apo: grey density, SLC26A7_I: blue density) around I1 and I2 sites.

The cryo-EM map quality around ion binding sites of SLC26A7 is shown. Additional density for iodide ions is explicit in SLC26A7_I map. The maps of SLC26A7_apo and SLC26A7_I are shown in $\sigma=2.8$ and 4.5, respectively.

Figure 2| Non-protein density in SLC26A7_apo map, SLC26A7_I map, and Difference map.

Non-protein density in SLC26A7_apo map (grey), SLC26A7_I map (green), and Difference map (blue) are shown, respectively. All maps are shown in $\sigma=5$. Only map of SLC26A7_apo of I2 site is shown in lower threshold of $\sigma=3$.

2. The authors claim that there are two separate transport pathways, with each passing through one substrate binding site. This is a bold claim as, at least in SLC26 members, no such observation has been made. It is a consensus note that SLC26 members go through transport cycles under the classic alternating access mechanism. The observed 2nd I⁻ binding site is most likely resides in the extracellular side of the only substrate translocation pathway.

Thank you for pointing out this question. We noticed there was a paper supposing the existence of other transport path in SLC26s (DOI: 10.1038/srep46619). However, there is indeed no robust evidence supporting the existence of two separate transport pathways. More structural and biochemical evidences are needed to claim the existence of 2nd transport path. We therefore move this part to the discussion section. Please refer to the update manuscripts.

3. The authors claim that the determined cryo-EM structures of SLC26A7 are both in the intermediate states. Are we talking about transporters or channels? From my understanding, it is mostly a transporter, but with some Cl⁻ channel activity (PMID: 32119864). If it's in the intermediate state, is the substrate binding pocket (between TM3 and TM10) accessible from both sides of the membrane? Such an intermediate state in transporters is also referred to as an occluded state that is close on both ends of the translocation pathway. The authors need to clarify the status of the structure more clearly.

This is a very interesting point of SLC26A7 structures. A well-known example for “transporter also channel” is GltPh (PMID: 33597752), in which the existence of separate ion permeation pathways is clearly identified. From the view of protein structure, SLC26A7 is a transporter as it exhibits classical UraA fold with Core and Gate domain in transmembrane region. It also behaves the elevator model in the transport cycles. From the view of transport dynamics, it behaves like a “channel” with rapid anion translocation speed so that electrophysiological measurements are suitable in detecting SLC26A7 transport activity (PMID: 32119864). We have added this citation in the manuscripts.

“Intermediate” state was reported in SLC26A9 (PMID: 31339488, PDB ID: 6RTF) and Prestin (PMID: 34695838, PDB ID: 7S9D). In both cases, the “intermediate” conformation is the transitional state between “Up” (outward) and “Down” (inward) state. In SLC26A7, inward-facing and outward-facing structure information is lacking. “Intermediate” state was adopted in our study because the highest conformational similarity was found between SLC26A7 and Prestin intermediate states (which is shown in main figure 4b in the manuscript).

The accessibility of substrate ion in I1 site was analyzed by pore radius (which is shown in supplementary figure 4c in the manuscript). Substrate in I1 site is not accessible to both sides. At extracellular side, the substrate was restricted by F68, I413, V190, and A105 with radius less than 1 Å. At cytosolic side, the substrate is restricted by G367, G419, and M366 with radius ~1.5 Å. The radius at the restriction sites are smaller than hydrated chloride radius (around 3 Å, DOI: 10.1016/j.seppur.2005.12.020). We conclude that the structure of SLC26A7 is captured in occluded state. We also modified the manuscripts accordingly.

4. What is the point of Fig. 4a? The authors performed such cluster analysis by including both experimental and predicted structures, which does not make too much sense. One compares either all available experimental models or only computationally predicted models.

The analysis was modified according to your suggestion. We have excluded the predicted structures, and included recent experimental structures of Pendrin and SLC26A2. There still exist 3 major groups, which are outward-facing, inward-facing, and intermediate. Only bacterial homolog SLC26Dg shows obvious difference with other structures. The intermediate group exhibits the highest heterogeneity. We have checked the structures of this class and found the heterogeneity comes from the extracellular loop of TM3-TM4 linkage. SLC26A7 apo and iodide loaded states are still classified in the intermediate states, confirming the results of ion stability analysis and pore radius measurements.

Minor concerns:

1. Numerous grammar errors need to be carefully corrected.

Thank you for pointing out this question. We have done the proof-reading of the paper, and have corrected the grammar errors.

2. Page 2: known SLC26 structures are not complete. Human SLC26A2 is missing. It's probably better to include structures of determined bacterial homologs, too.

We have included SLC26A2 and bacterial homologs according to your suggestion in this version of manuscript.

3. The model building can be further improved. For example, the backbone of multiple residues is clearly wrong, and the Ramachandran plot can be improved.

We have modified the model building as suggested.

4. Supp fig 2e legend is incorrect.

Sorry for the mistake. We have corrected this mistake in the updated manuscript.

Reviewer #2 (Remarks to the Author):

The manuscript by Li and colleagues reported the structures of human SLC26A7 in apo and iodide binding states. Like the structure of Pendrin, the structure of SLC26A7 forms a dimer and adopts an UraA-fold. The TMD can be divided into gate and core domains. Based on structural analysis, two putative ion binding sites were identified. One canonical ion binding site is located in the pocket formed by TM3 and TM10 highly which is conserved in UraA-folded transporters. What is potentially new is that the iodide-loaded structure also potentially revealed a non-canonical ion binding site, which is located in the pocket formed by TM5, TM13 and TM14 in the gate domain. Furthermore, key residues involved in ion coordination were studied using electrophysiological assays. Additional evidence is required as outlined in my comments to support several of the conclusions.

Thank you for reviewing our manuscripts. The point-to-point answers are attached to each of the questions below.

Major

1) In line 90-91, the authors stated that “Several cholesterol-like molecules are presented around the TMDs (Supplementary Fig. 2a)”. The densities are in the peripheral regions, where the resolution is low. How can the authors distinguish the densities are not part of the detergent micelle.

Detergent molecules are attached to membrane protein without specificity. The corresponding density is weak in cryo-EM map after rounds of 3D classification. On the contrary, the phospholipids or cholesterol which survived the purification process attach to membrane protein more tightly than detergents. Although the densities are in the peripheral regions, they persist in high threshold ($\sigma=4.8$), indicating a tight binding mode. Similar cholesterol-like molecules binding in the peripheral regions were also observed in the structure of Prestin (PMID: 34390643).

2) In line 119-121, the authors said “The cryo-EM map of SLC26A7 with iodide shows additional non-protein densities in the long cleft between core and gate domain when compared with SLC26A7 apo state. Iodide ions were therefore built in these sites, namely I1 and I2, respectively”. In addition, a water molecule was identified in I1 site. However, the quality of the map was not shown. The authors should calculate the difference map. The omitted density can be used to evaluate the signal of the built ions. Thank you for pointing out this question. The map quality around I1 and I2 sites are shown as suggested (Figure 1). Density of the two iodide ions can clearly be visualized in the comparison (Figure 1 and 2). We also include this information in the revision (Fig. 2a, and first paragraph of section “Iodide coordination in SLC26A7”).

3) In line 155, the authors claimed that “Both pores share the same entry and exist”. But shown in Fig.3b, pore1 and pore2 have a different entry and exit. This needs clarification.

Thank you for pointing out this question. We have throughout reviewed the manuscripts and our structures, and realized that solid evidence is still needed to conclude the existence of the second ion permeation path. Moreover, Pore1 and Pore2 was calculated by HOLE based on the coordination of L71 and A453 respectively. In SLC26A7, Pore1 and Pore2 share the same entry and exit (Supplementary figure 4b,d). We also noticed that the results vary when different residues are selected in the calculation. We therefore move this part to discussion section.

4) In line 240-242, “minor deformation of the core domain provides subtle adjustment of substrate recognition and coordination during transport cycles.”. The statement needs more evidence to support it, to help address in greater detail the structural detail/changes in the transport cycle.

Thank you for pointing out this question. As structures of SLC26A7 in different transport states are lacking, we are not able to compare substrate binding pocket change among different conformational states of SLC26A7. Considering the structural similarity, we compared the substrate binding pocket of Prestin during transport cycle

(Figure 3). Structures of Prestin in different states are involved, which includes 7S9B (down), 7S9C (down), 7LH2 (intermediate), and 7S9D (intermediate). Obvious misalignments of TM3 and TM10 are detected in these structures. Conformational differences of P136, A138, S398, and L397 change the structure of substrate binding pocket. Specially, conformational change of L397 in 7S9D takes up salicylate binding site. Therefore, there is the possibility that core domain deformation changes substrate binding pocket environment during transport cycle, which could provide different binding affinity to the substrates facilitating substrate recruitment and release.

Figure 3| Structure alignment of Prestin in different states

Intermediate and down states structures of Prestin are aligned. While TM1 remains mostly the same, TM3, TM10, and TM8 behave obviously deformation. Enlarged figures in the left zooms in substrate binding area, side-chain conformational change in P136, S398, and L397 affect the environment of substrate coordination. L397 of Prestin (Intermediate,7S9D) is in clash with salicylate in structures of Prestin (Intermediate, 7LH2).

The comparison between SLC26A7 and Prestin was included in the updated manuscripts as Supplementary figure 3e and f. When compared with Prestin (down), SLC26A7 shows obvious deformation both in backbone and residues of I1 pocket (Supplementary figure 3e). When aligned to Prestin (intermediate or up), side-chain conformational change will potentially interfere substrate binding (Supplementary figure 3f).

We noticed that more experimental evidences are needed to make the solid conclusion of the physiological importance of core domain deformation in SLC26s. We therefore

change the sentence as “Therefore, deformation of core domain in SLC26s can provide subtle adjustment of substrate recognition and coordination during transport cycles”.

5) Although the authors showed that density observed at I2 site is likely to be I⁻ by MD, there is no proof that the density at I1 is I⁻. It could also be Cl⁻. The possibility of endogenous Cl⁻ binding during protein expression and purification cannot be excluded. Thank you for pointing out this question. As Cl⁻ and I⁻ are halogen ion with high chemical similarity, it is reasonable to be Cl⁻ at I2 site. However, Cl⁻ is not always presented in the structures of SLC26s, even in high Cl⁻ concentration buffers (ranging from 150 to 360 mM NaCl, PMID: 34695838 and 32818062, 31339488). The comparison of the cryo-EM maps of SLC26A7_I and SLC26A7_apo indicates that non-protein cryo-EM densities are presented in I1 and I2 sites (Figure. 1). The higher atomic weight of iodine also supports the enhanced map signal. Taken together the MD simulation results, Iodide ions are therefore built in the model. We have add the information in manuscript as “The iodide ions were built in the SLC26A7_I model based on the following clues. Firstly, cryo-EM map comparison indicates strong additional signals in I1 and I2 sites of SLC26A7_I map (Fig. 3a). The only difference in protein sample preparation is sodium iodide addition. It is iodide binding that induces the signal enhancement. Secondary, SLC26A7_apo shows weak density of suspected Cl⁻ in I1 and I2 sites when the map threshold is lowered down, which indicates Cl⁻ binding. But it is not enough for a robust model-building. Iodide ion has higher atom weight and exhibit better signal in cryo-EM data than chloride ion. Therefore, iodide provide rationality for the enhanced map density. Thirdly, MD simulation results support that while both Cl⁻ and I⁻ are stable in I1 site, only I⁻ is stable in I2 site. Yet the binding affinity of Cl⁻ and I⁻ to SLC26A7 needs to be measured for ion species determination.”

6) In some UraA-folded transporters, TM8 is also part of the canonical binding pocket. Are there any residues located in TM8 involved in ion coordination in SLC26A7?

The TM8 residue involved in substrate coordination was observed in human Prestin structure (PDB ID: 7LGU, PMID: 34390643). In this structure, V353 is one of the

hydrophobic residues surrounding substrate Cl⁻. In SLC26A7, V353 was replaced by A322. However, the distance between A322 and iodide in I1 is around 8 Å, which is too far to set up a specific coordination. So, there is no clue of A322 participating in substrate binding in SLC26A7. But we involved A322 (V353 in Prestin) in structure alignments in Supplementary figure 3e and f so as to make it more comparable.

Minor

1) In Fig.1, add labels to transmembrane helices of the model.

Missing of the labels would lead to difficulty in structure analysis. We added the top view of transmembrane domain in Fig. 1b. Besides, enlarged view of gate and core domain separately are presented in Fig. 1c. The numbers of transmembrane helices are labeled in the updated figures as suggested.

2) In line 230-231, AlphaFold 3 prediction of chloride bound SLC26A7 was performed. What was the AlphaFold 3 prediction of iodide bound SLC26A7 since iodide was built in I2 in the model?

Currently ion species are restricted in AlphaFold 3. Only chloride is available in structure prediction by AlphaFold 3.

3) Add SEC profile and gel for sample purification in Supplementary Fig.1. Specify the buffer conditions used for SEC in protein expression and purification section.

We add SEC profile and gel as suggested in Supplementary Fig. 1a. The buffer for SEC is 25 mM Tris, pH8.0, 150 mM NaCl, and 0.02% GDN. We have modified the sentence in protein expression and purification section as “Then the protein was further purified by size exclusion chromatography (SEC) (Superose 6 Increase 10/300 GL, GE Healthcare) in solution buffer: 25 mM Tris, pH8.0, 150 mM NaCl, and 0.02% GDN.”.

Reviewer #3 (Remarks to the Author):

Thank you for your effort! The listed questions were answered, and the manuscripts were modified accordingly.

Response to reviewers

REVIEWER COMMENTS

Reviewer #2 (Remarks to the Author):

I have carefully reviewed the revised manuscript and the authors responses. I have no further comments.

Thank you for reviewing our manuscripts.

Reviewer #3 (Remarks to the Author):

Thank you for reviewing our manuscripts.

Reviewer #4 (Remarks to the Author):

This is a technical review for the MD simulations:

1) The phrase "Simulations was carried out every 100 ps with duration of 100 ns with 3 replications" is unclear and needs clarification and editing. Specifically, what do the authors mean by "Simulations was carried out every 100 ps"?

We have corrected the sentence to make it clear.

"The simulations consisted of three independent 100 ns replicates, with conformational sampling performed every 100 ps."

2) In Figure 3a, it would be helpful if the authors could explain what each line, represented in different colors, stands for.

We have added the explanation in the legend to make the figure clear.

"a. MD simulation estimating the ion species and binding stability in I1 and I2 sites indicates that while both Cl⁻ and I⁻ are stable in I1 site, I2 site is more stable for I⁻ binding and instable for Cl⁻. In

each panel, the six different colored lines represent the distance between ion and each site in the dimer system across different replicate simulations.”

3) Can the authors mention and cite any molecular dynamics (MD) work where a time length of 100 ns was used to assess the structural stability of ion binding sites in channel proteins?

We appreciate the reviewer’s suggestion. We have mentioned in Methods section and added the corresponding citations.

“To assess the ion binding stability using the conventional MD simulations¹⁻³, the 100 ns production simulations with three replicas were performed to sample the snapshots every 100 ps.”

4) The authors should clarify which parameters were used for the I⁻ and Cl⁻ ions. The cited reference in the Supplementary Material points to the parametrization of cations (supplementary Ref. 23).

Thank you for the comment. We have clarified the parameters of I⁻ and Cl⁻ ions used in MD simulations.

“The force field parameters for Cl⁻ are directly adopted from the CHARMM force field⁴. For I⁻, the parameters were derived based on the reported difference in hydration free energy between I⁻ and Cl⁻ ions⁵. The parameters for Cl⁻ are $R_{min}/2=2.27$ and $\epsilon=-0.15$, while those for I⁻ are $R_{min}/2=2.76$ and $\epsilon=-0.184$.”

5) Related to Figure 3a, which site is the most prone to ion instability? The colored lines should perhaps report on this. Moreover, could it be possible to have a system with Cl⁻/I⁻ bound?

Thank you for the comment. We have updated the legend to provide greater clarity regarding the colored lines.

“a. MD simulation estimating the ion species and binding stability in I1 and I2 sites indicates that while both Cl⁻ and I⁻ are stable in I1 site, I2 site is more stable for I⁻ binding and instable for Cl⁻. In each panel, the six different colored lines represent the distance between ion and each site in the dimer system across different replicate simulations.”

MD simulation was set up to help identification of possible ion species which confirmed the results of structure analysis (Ion densities are shown in Fig. 2a). Analysis of a mixed Cl⁻/I⁻ bound system would make the system unnecessarily complicated. We therefore not considered it in our study.

6) Detailed information should be provided on how HOLE was used to build the path collective variable (CV) mapped with Umbrella Sampling.

Thank you for the suggestion. Detailed information on the use of HOLE to construct the CV has been added.

“Specifically, the collective variables were defined as the central path along the pore, calculated using the HOLE program⁶. HOLE identifies the pore's central axis by determining the pathway of

maximum radius within the channel, ensuring that it accurately reflects the ion conduction pathway. Then, two transport paths for Cl⁻ and I⁻ were obtained to define the reaction coordinate for umbrella sampling simulations.”

7) Error bars should be included for the potential of mean force (PMF) profiles in Supplementary Figure 4.

Thank you for the suggestion. We have added the error bars in the Supplementary Figure 4e,f.

8) The X-axis label of the PMF (Supplementary Figure 4e,f), which simply reports "in" and "out," should indicate which window each portion of the PMF belongs to. Ideally, for comparison with the tunnel features shown in Supplementary Figure 4c, the free energy profile should also be plotted as a function of the "Distance along the permeation Path."

Thank you for the suggestion. We have modified the Supplementary Figure 4e,f with the required information added. We also modified the figure legends accordingly.

9) It should be clarified which portion of the PMF corresponds to the "ion binding" site. Again, the X-axis of the PMF should report the distance along the permeation path and the corresponding Umbrella sampling window. This will facilitate following the final part of the manuscript's discussion.

Thank you for the suggestion. We have modified the Supplementary Figure 4e,f. Ion binding sites are labeled in HOLE and PMF analysis.

10) References should be provided when citing Umbrella Sampling.

Thank you for the suggestion. We have added the citation in the updated manuscripts.

11) For the sake of clarity and to allow the community to build on the simulations performed by the authors, PLUMED input files and corresponding simulation input files should be shared via the PLUMED-NEST.

Thank you for the suggestion. We have shared the corresponding files via PLUMED-NEST and claimed in Data availability.

*“The initial data and PLUMED input files required in MD simulations to reproduce the results reported in this paper are available on PLUMED-NEST (www.plumed-nest.org), the public repository of the PLUMED consortium, as *plumID:24.034*.”*

12) The manuscript, including the main text and the Methods section in the Supplementary Material, would benefit from careful proofreading and editing.

Thank you for the suggestion. We have performed the careful proofreading and editing.

References:

- 1 Khafizov, K. *et al.* Investigation of the sodium-binding sites in the
sodium-coupled betaine transporter BetP. *Proceedings of the National
Academy of Sciences* **109**, E3035–E3044 (2012).
- 2 Perez, C. *et al.* Substrate-bound outward-open state of the betaine
transporter BetP provides insights into Na⁺ coupling. *Nature
Communications* **5**, 4231 (2014).
- 3 Guskov, A., Jensen, S., Faustino, I., Marrink, S. J. & Slotboom, D. J.
Coupled binding mechanism of three sodium ions and aspartate in the
glutamate transporter homologue GltTk. *Nature Communications* **7**, 13420
(2016).
- 4 Beglov, D. & Roux, B. Finite representation of an infinite bulk system:
Solvent boundary potential for computer simulations. *The Journal of
chemical physics* **100**, 9050–9063 (1994).
- 5 Won, Y. Force field for monovalent, divalent, and trivalent cations
developed under the solvent boundary potential. *J Phys Chem A* **116**,
11763–11767, doi:10.1021/jp309150r (2012).
- 6 Smart, O. S., Neduvélil, J. G., Wang, X., Wallace, B. A. & Sansom, M. S.
HOLE: a program for the analysis of the pore dimensions of ion channel
structural models. *J Mol Graph* **14**, 354–360, 376 (1996).

We are thankful for the precious suggestions and the devotion of each reviewer provided for the manuscript! Here are the response for the reviewers' comments.

REVIEWERS' COMMENTS

Reviewer #2 (Remarks to the Author):

I have carefully reviewed the revised manuscript and the authors responses. I have no further comments.

Thank you for your devotion in the review.

Reviewer #3 (Remarks to the Author):

Thank you for your devotion in the review.

Reviewer #4 (Remarks to the Author):

I have carefully read the revised manuscript, and the authors have sufficiently addressed my concerns. I only have a final suggestion regarding the wording of the sentence in the legend of Figure 3: "The simulations consisted of three independent 100 ns replicates, with conformational sampling performed every 100 ps."

I assume the authors intend to convey that sampling was continuous throughout the 100 ns simulations, with snapshots of the system saved at 100 ps intervals. However, the current phrasing — "with conformational sampling performed every 100 ps" — could suggest that sampling only occurred at those specific intervals, rather than continuously.

To avoid this potential confusion, I suggest rephrasing it to something like: "with snapshots of the system saved every 100 ps for analysis."

This would clarify that data points were recorded at regular intervals.

Thank you for the suggestion. We have rewritten the sentence as "The simulations consisted of three independent 100 ns replicates, with snapshots of the system saved every 100 ps for analysis." in the figure legend of figure 3.